# Risk factors and etiology of neonatal sepsis after hospital delivery: A case-control study in a tertiary care hospital of Rajshahi, Bangladesh

**Md. Abdur Rafi** [1], **M. Morsed Zaman Miah**[1], **Md. Abdul Wadood**[2], **Md. Golam Hossain** [3]*

1 Rajshahi Medical College, Rajshahi, Bangladesh, 2 Medical Centre, University of Rajshahi, Rajshahi, Bangladesh, 3 Health Research Group, Department of Statistics, University of Rajshahi, Rajshahi, Bangladesh

* hossain95@yahoo.com

## Abstract

### Background

Sepsis is one of the major causes of neonatal death worldwide as well as in Bangladesh. The objective of the present study was to identify the risk factors and causative organisms of neonatal sepsis after delivery in a tertiary care hospital, Bangladesh.

### Methods

This was a case-control study conducted in the neonatal ward of Rajshahi Medical College Hospital (RMCH), a 1000-bed tertiary hospital situated in Rajshahi, Bangladesh. Neonates diagnosed as neonatal sepsis by clinical and laboratory parameters were included as cases in this study. Admitted neonates unsuspected or undiagnosed for sepsis were considered as controls. Maternal and neonatal information and their laboratory reports were collected and analyzed. Both bivariate and multiple logistic regression models were used to identify the risk factors of neonatal sepsis.

### Results

A total of 91 cases and 193 controls were included in the study. Maternal history of urinary tract infection (UTI) during the third trimester of pregnancy (aOR 2.75, 95% CI: 1.04–7.23, p <0.05), premature birth (aOR 2.77, 95% CI: 1.08–7.13, p <0.05) and APGAR score <7 at five minutes (aOR 2.58, 95% CI: 1.04–6.39, p <0.05) were associated with onset of neonatal sepsis in multiple logistic regression model. All these factors were also associated with developing early-onset neonatal sepsis, while maternal UTI and male sex of neonates were associated with developing late-onset neonatal sepsis. *Escherichia coli* (40.7%), *Staphylococcus aureus* (27.5%), and *Klebsiella pneumoniae* (18.7%) were the commonly isolated organisms causing neonatal sepsis. All these organisms were highly resistant to common antibiotics like amoxicillin, cephalosporins, aminoglycosides and quinolones.

**Data Availability Statement:** All relevant data are within the paper and its Supporting Information files.

**Funding:** The authors received no specific funding for this work.

**Competing interests:** The authors have declared that no competing interests exist.

Carbapenemase group of drugs along with amikacin, nitrofurantoin and linezolid were the most sensitive drugs.

## Conclusions

Strengthening the existing facility for antenatal screening for early diagnosis and treatment of maternal infection during pregnancy as well as identifying high-risk pregnancy for adequate perinatal management is necessary to prevent neonatal sepsis-related morbidity and mortality. Rational use of antibiotics according to local epidemiology and culture and sensitivity reports may minimize the increasing hazards of antibiotic resistance.

## Introduction

Neonatal sepsis is a significant but neglected public health concern, especially in the lower and middle-income countries of sub-Saharan Africa and South-East Asia. Consequently, despite a decreasing trend in global neonatal mortality during the last two decades, the rate of reduction of sepsis-specific mortality has been much slower compared to that of other causes like premature birth or intrapartum complications in these regions [1]. More than six million neonates suffer from severe infections and sepsis annually in these regions [2]. It contributes to almost one-quarter of global neonatal deaths per year [3]. Bangladesh is a developing country in the South-East Asian region where sepsis contributes to almost 37% of all neonatal deaths [4].

Neonatal sepsis denotes a systemic inflammatory response syndrome in the presence of or as a result of suspected or proven infection in a neonate (within the first 28 days of life) [5, 6]. According to the age of onset, neonatal sepsis is divided into two classes: (i) early-onset neonatal sepsis (EONS) and (ii) late-onset neonatal sepsis (LONS). Usually, the onset of sepsis within the first 72 hours of life is referred to as EONS, while the onset of sepsis after 72 hours but within the first 28 days of life is referred to as LONS. It is anticipated that organisms acquired before and during delivery (or maternal-fetal infection) are mainly responsible for EONS, while organisms acquired after delivery from the environment (nosocomial or community sources) are responsible for LONS [7]. Case fatality of neonatal sepsis is high and many of the surviving neonates suffer from poor long-term neurodevelopmental outcomes as a consequence of CNS involvement, septic shock, or hypoxemia secondary to severe parenchymal lung disease [8]. Bacterial pathogens such as *Klebsiella pneumoniae*, *Staphylococcus aureus*, and coagulase-negative *Staphylococcus* are the most common causes of neonatal sepsis in developing countries [9]. These organisms are highly resistant to commonly used antibiotics, which makes them challenging to treat [10–12]. Moreover, early diagnosis and treatment of newborns with infection are unsatisfactory in resource-poor settings, which contributes to the high neonatal mortality due to sepsis [13].

There are epidemiological differences in the incidence, risk factors, pattern, and antimicrobial sensitivities of pathogens and mortality of neonatal sepsis among different regions and countries in the world [14]. Specific strategies suitable for specific countries to prevent and treat neonatal sepsis are needed to accelerate the progress of preventing neonatal morbidity and mortality. Identification of risk factors and early diagnosis and institution of therapy according to local epidemiology and antimicrobial resistance pattern can improve neonatal survival. There is a lack of evidence on risk factors and common causative agents with their sensitivity patterns of neonatal sepsis in Bangladesh. A case-control study conducted in Bangladesh focused on community-acquired LONS showed that underweight, admission in the

winter season, primiparity, and home delivery were associated with LONS [15]. Another community-based study reporting neonatal infection within the first nine days of life demonstrated that history of child death in the family, large family, home delivery, unclean cord care, multiple birth, low birth weight, and perinatal asphyxia were associated with neonatal infection [16]. Some other studies reported that neonatal sepsis was associated with low birth weight, preterm neonates, meconium-stained liquor, and prolonged rupture of membrane [17, 18]. Gram-negative organisms predominated with *Escherichia coli* are the commonest causative agents, though this study is not robust enough to verify the risk factors of neonatal sepsis due to its cross-sectional nature [17]. However, these studies did not represent the scenario of the whole country as there were socio-economic and demographic disparities across the country. Moreover, most of the studies were community-based and authors considered births at both home and hospital.

The present study, therefore, aimed to determine the risk factors and etiology of neonatal sepsis among the neonates delivered in a tertiary care hospital in northern Bangladesh.

## Methods

### Study design and setting

This was a case-control study conducted in the neonatal ward of Rajshahi Medical College Hospital (RMCH) from January to December 2019. RMCH is a 1000-bed tertiary care hospital situated in Rajshahi city and it is the main referral medical institution in northern Bangladesh. This hospital has a 50-bed neonatal ward with a neonatal intensive care unit (NICU) facility.

### Study population and sampling

The study population was all the neonates aged 0–28 days who were admitted to the neonatal ward of RMCH and born in this hospital either by vaginal delivery or cesarean section during the study period. A two population proportion formula (using open Epi version 2.3.1) was used to calculate the adequate sample size for the study by considering the proportion of mothers with a history of UTI during the third trimester (proved by a positive urine culture report or the presence of >5 pus cells/HPF in the microscopic test of urine with a history of clinical symptoms of UTI such as fever, burning sensation in micturition, etc., for those whose urine culture report was not available), among the controls of 13.5% (one of the main exposure variables), which was estimated from a previous study [19], 95% confidence interval, 80% power of the study control to case ratio of 2:1 to detect an estimated odds ratio of 2.5 and 10% non-response rate. Accordingly, 93 cases and 186 controls (a total sample size of 279) were enough. Cases and controls were selected using a proportional convenient sampling method according to inclusion and exclusion criteria. The distribution of cases and controls is shown in the Fig 1.

### Inclusion and exclusion criteria

Neonates diagnosed for neonatal sepsis according to established clinical and hematological criteria of IMNCI (Integrated Management of Neonatal and Childhood Illness) and evidence of positive blood culture results were included as cases in this study. Neonates who were not suspected or diagnosed for neonatal sepsis and who were born in the obstetrics department and/ or admitted to the neonatal ward of RMCH due to other indications such as low birth weight, neonatal jaundice, diarrhea, etc. during the study period were included as controls. Controls were matched for age with cases. Neonates who were born outside RMCH or clinically suspected as sepsis but not confirmed by hematological test and positive blood culture were excluded from the study.

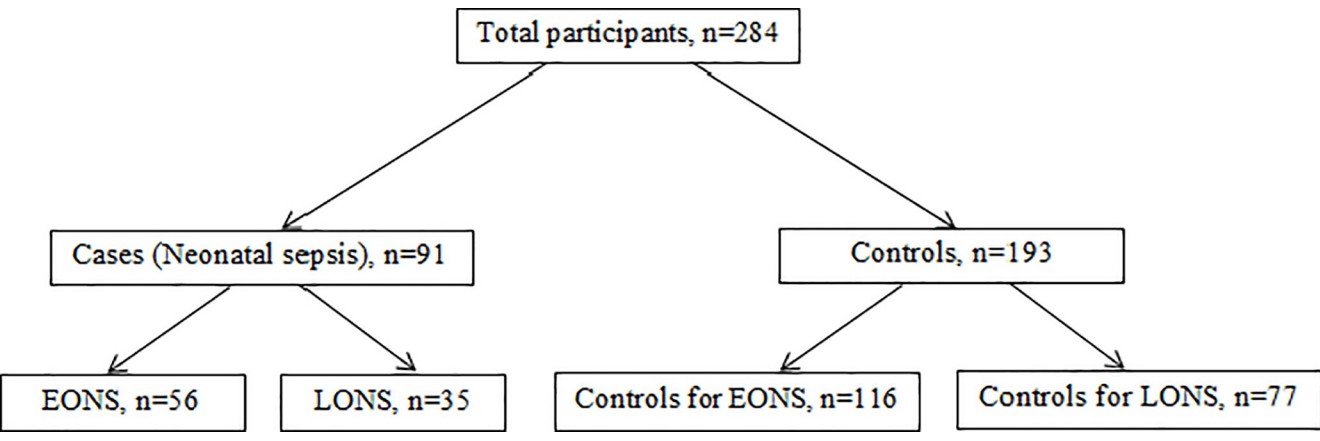

**Fig 1. Distribution of cases and controls.**

### Diagnosis of neonatal sepsis

Neonatal sepsis includes a range of systemic infections of the newborn, such as septicemia, meningitis, pneumonia, arthritis, osteomyelitis, etc. [6]. Among all these infections, septicemia, defined as generalized bacterial infection documented by a positive blood culture within the first 28 days of life, contributes to a large portion of neonatal morbidity and mortality due to sepsis [20]. We considered neonates with positive blood culture results as sepsis cases, as the facility of culture for other specimens was absent in the study center. According to the international pediatric sepsis consensus conference, neonatal sepsis was diagnosed by the presence of one or more of the established clinical features [either of fever ($>37.5°C$) or hypothermia ($<35.5°C$), fast breathing ($>60$ breaths per minute), severe chest indrawing, not feeding well, the movement only when stimulated, convulsion, lethargic or unconscious] and confirmed by the presence of two of the hematological criteria (total leukocyte count $<4000$ or $>12000$ cells/mm$^3$, absolute neutrophil count $<1500$ cells/mm$^3$ or $>7500$ cells/mm$^3$, erythrocyte sedimentation rate (ESR) $>15$ in the first hour and platelet count $<150$ or $>440$ cells/m$^3$) along with a positive blood culture test [5]. Early-onset neonatal sepsis (EONS) was defined as sepsis occurring within the first 72 hours of life, and that occurring after 72 hours and within 28 days of life was defined as late-onset neonatal sepsis (LONS) [19].

### Blood culture and antimicrobial susceptibility testing

The blood sample was collected from clinically suspected patients by trained pediatric nurses and sent to the microbiology lab of RMCH for culture. 1 ml of blood was collected from each patient and directly inoculated into a pediatric FAN blood culture bottle. If a collection of 1 ml of blood was not possible or contraindicated by the consultant neonatologist, especially in low birth weight premature neonates due to their weight and total blood body volume, 0.5 ml of blood was collected. As blood of a lower volume than 0.5 ml significantly reduces the sensitivity [21], neonates from whom collection of 0.5 ml of blood was not clinically justified by the consultant neonatologist were excluded from the study. Collected samples were incubated in the BACT/Alert machine for up to 5 days. The BACT/Alert system is reported as highly sensitive with a positivity index of 0.94 for the gram-positive organism (*S. epidermidis*) and 0.95 for the gram-negative organism (*E. coli*) [21]. For identification of organisms, positive culture samples were directly inoculated onto MacConkey (MC) agar, chocolate agar, and blood agar (5% sheep blood) plates. MC plates were then incubated at 35°C in aerobic conditions.

Chocolate and blood agar plates were incubated at 35°C in microaerophilic conditions (containing 5% $CO_2$). Isolated organisms were identified using standard laboratory procedures. API 20E identification strips (bioMérieux, France) were used for further identification. Antimicrobial susceptibility test (AST) was carried out by Kirby–Bauer disc diffusion method [22] and susceptibility patterns were determined following CLSI guidelines [23]. Antimicrobial susceptibility was tested for a panel of antibiotics (Oxoid, UK). *E. coli* ATCC 25922 and *Pseudomonas aeruginosa* ATCC 27853 were used as quality control strains. Non-susceptibility to at least one agent in three or more antimicrobial categories was defined as multidrug resistance (MDR) [24].

## Data collection procedure

Data related to maternal and neonatal risk factors as well as laboratory parameters and blood culture reports were extracted from medical records of the mother during delivery and admitted neonates using a structured checklist. In the case of any missing information about maternal and neonatal risk factors, the mother of the neonate was interviewed by two trained nurses. Detailed socio-demographic information was not collected due to our limited resources.

## Outcome variable

In this study, neonatal sepsis was the primary outcome variable. This was further classified into two classes: (i) early-onset neonatal sepsis (EONS) defined as the onset of sepsis within the first 72 hours of life, and (ii) late-onset neonatal sepsis, defined as the onset of sepsis after the first 72 hours but within 28 days of life.

## Independent variables

Independent variables were: maternal variables such as maternal age, maternal UTI during the third trimester of pregnancy, prolonged labor, premature rupture of membrane, and mode of delivery; and neonatal variables such as the age of neonate, sex of neonate, prematurity (gestational age), weight at birth, and APGAR score at 5 minutes. These potential risk factors were selected from the evidence of previous studies [19, 25, 26].

## Ethics approval and consent to participate

Ethical approval for this study was obtained from the Ethical Review Committee of Rajshahi Medical College [Ref. RMC/ERC/2017-2019/199/179]. Informed written consent was obtained from the mother of the neonate before enrollment in the study.

## Statistical analysis

All statistical analyses were carried out using SPSS (IBM version 24.0). Statistical analysis was done: (i) to determine the risk factors of neonatal sepsis including all cases and controls and (ii) to determine the risk factors of early and late-onset neonatal sepsis including the cases aged 0 to 72 hours and 72 hours to 28 days and their age-matched controls respectively. The chi-square test was carried out to determine differences among groups concerning the outcome variables. Both bivariate and multiple logistic regression models were used to generate crude odds ratios (cOR) and adjusted odds ratios (aOR) with 95% confidence intervals (CI) for significance testing. Variables that achieved p-value <0.05 in bivariate analysis were entered into the multiple logistic regression models.

# Results

## Characteristics of the participants

A total number of 284 participants were included in the study, 91 diagnosed as neonatal sepsis (cases), and 193 diagnosed as no sepsis (controls). Among cases, 56 (61.5%) and 35 (38.5%) were diagnosed as EONS and LONS respectively. Almost half of the neonates of both cases and controls were female. 65.5% of the mothers of neonates were of 18–35 years; 14% of them had a history of UTI during the third trimester of pregnancy (24% among cases, 9% among controls). 16% of mothers had prolonged labor (24% among cases and 13% among controls), and 19.4% of them had premature rupture of membrane (27.5% among cases and 15.5% among controls). A total of 38.7% of neonates were delivered by cesarean section (49.5% among cases and 33.7% among controls). Prematurity, low birth weight, and APGAR score <7 at 5 minutes were more prevalent among cases (46% vs 18.7%, 54% vs 25.4%, and 31% vs 11.4% respectively among cases and controls). The chi-square test demonstrated that maternal UTI during 3rd trimester, prolonged labor, premature rupture of membrane, mode of delivery, prematurity, weight at birth, and APGAR score at 5 minutes were significantly associated with neonatal sepsis (Table 1).

## Risk factors of neonatal sepsis

Multiple binary logistic regression model demonstrated that mothers having UTI during the 3rd trimester had a 2.75-fold greater risk of neonatal sepsis than their counterparts (aOR = 2.75; 95% CI: 1.04–7.23; p<0.05). Premature neonates were at 2.77 times higher risk for getting sepsis than neonates delivered in time (aOR = 2.77, 95% CI: 1.08–7.13; p<0.05). Neonates having a lower APGAR score (<7) at 5 minutes were more likely to get sepsis than neonates with APGAR score of 7 and above (aOR = 2.57, 95% CI: 1.02–6.43; p<0.05).

**Table 1. Maternal and neonatal characteristics of cases and control.**

| Variables | Category | Total, N (%) | Case, N (%) | Control, N (%) | p-value |
|---|---|---|---|---|---|
| Maternal age (year) | <18 | 60 (21.1) | 21 (23.1) | 39 (20.2) | 0.783 |
| | 18 to 35 | 186 (65.5) | 57 (62.6) | 129 (66.8) | |
| | >35 | 38 (13.4) | 13 (14.3) | 25 (13.0) | |
| Maternal UTI during 3rd trimester | Yes | 40 (14.1) | 22 (24.2) | 18 (9.3) | 0.001 |
| | No | 244 (85.9) | 69 (75.8) | 175 (90.7) | |
| Prolonged labor | Yes | 47 (16.5) | 22 (24.2) | 25 (13.0) | 0.018 |
| | No | 237 (83.5) | 69 (75.8) | 168 (87.0) | |
| Premature rupture of membrane | Yes | 55 (19.4) | 25 (27.5) | 30 (15.5) | 0.018 |
| | No | 229 (80.6) | 66 (72.5) | 163 (84.5) | |
| Mode of delivery | Caesarean | 110 (38.7) | 45 (49.5) | 65 (33.7) | 0.011 |
| | Vaginal | 174 (61.3) | 46 (50.5) | 128 (66.3) | |
| Age of neonate (day) | 0 to 3 | 172 (60.6) | 56 (61.5) | 116 (60.1) | 0.817 |
| | 3or 4–28 | 112 (39.4) | 35 (38.5) | 77 (39.9) | |
| Sex of neonate | Female | 151 (53.2) | 48 (52.7) | 103 (53.4) | 0.922 |
| | Male | 133 (46.8) | 43 (47.3) | 90 (46.6) | |
| Prematurity | Yes (<37 weeks) | 78 (27.5) | 42 (46.2) | 36 (18.7) | 0.001 |
| | No (≥37 weeks | 206 (72.5) | 49 (53.8) | 157 (81.3) | |
| Weight at birth (gram) | Low, <2500 | 98 (34.5) | 49 (53.8) | 49 (25.4) | 0.001 |
| | Normal, ≥2500 | 186 (65.5) | 42 (46.2) | 144 (74.6) | |
| APGAR score at 5 minutes | <7 | 50 (17.6) | 28 (30.8) | 22 (11.4) | 0.001 |
| | ≥7 | 234 (82.4) | 63 (69.2) | 171 (88.6) | |

**Table 2. Bivariate and multiple logistic regressions for risk factors of neonatal sepsis.**

| Variables | Category | Case, N (%) | Control, N (%) | cOR (95% CI) | aOR (95% CI) |
|---|---|---|---|---|---|
| Maternal age (year) | <18 | 21 (23.1) | 39 (20.2) | 1.04 (0.44–2.43) | |
| | 18 to 35 | 57 (62.6) | 129 (66.8) | 0.85 (0.41–1.78) | |
| | >35 | 13 (14.3) | 25 (13.0) | | |
| Maternal UTI during 3rd trimester | Yes | 22 (24.2) | 18 (9.3) | 3.10 (1.57–6.13)* | 2.75 (1.04–7.23)* |
| | No | 69 (75.8) | 175 (90.7) | | |
| Prolong labor | Yes | 22 (24.2) | 25 (13.0) | 2.14 (1.13–4.06)* | 1.89 (0.70–5.08) |
| | No | 69 (75.8) | 168 (87.0) | | |
| Premature rupture of membrane | Yes | 25 (27.5) | 30 (15.5) | 2.06 (1.13–3.76)* | 1.44 (0.60–3.45) |
| | No | 66 (72.5) | 163 (84.5) | | |
| Mode of delivery | Caesarean | 45 (49.5) | 65 (33.7) | 1.93 (1.16–3.20)* | 1.47 (0.67–3.22) |
| | Vaginal | 46 (50.5) | 128 (66.3) | | |
| Age of neonate (day) | 0 to 3 | 56 (61.5) | 116 (60.1) | 1.06 (0.64–1.77) | |
| | 3–28 days | 35 (38.5) | 77 (39.9) | | |
| Sex of neonate | Female | 48 (52.7) | 103 (53.4) | 0.98 (0.59–1.61) | |
| | Male | 43 (47.3) | 90 (46.6) | | |
| Prematurity | Yes (<37 weeks) | 42 (46.2) | 36 (18.7) | 3.74 (2.16–6.47)* | 2.77 (1.08–7.13)* |
| | No (≥37 weeks) | 49 (53.8) | 157 (81.3) | | |
| Weight at birth (gram) | Low, <2500 | 49 (53.8) | 49 (25.4) | 3.43 (2.03–5.79)* | 1.27 (0.50–3.26) |
| | Normal, ≥2500 | 42 (46.2) | 144 (74.6) | | |
| APGAR score at 5 minutes | <7 | 28 (30.8) | 22 (11.4) | 3.46 (1.84–6.48)* | 2.57 (1.02–6.43)* |
| | ≥7 | 63 (69.2) | 171 (88.6) | | |

N.B.

*p-value <0.05, CI: Confidence interval, cOR: crude odds ratio, aOR: Adjusted odds ratio.

However, bivariate logistic model showed that prolonged labor (cOR = 2.14, 95% CI: 1.13–4.06; p<0.05), premature rupture of membrane (cOR =: 2.06, 95% CI: 1.13–3.76; p<0.05), mode of delivery (cOR = 1.93, 95% CI: 1.16–3.20; p<0.05), and weight at birth (cOR = 3.43, 95% CI: 2.03–3.26; p<0.05) were the most influential factors of neonatal sepsis (Table 2).

## Risk factors for early and late-onset neonatal sepsis

Tables 3 and 4 show the risk factors of early and late-onset neonatal sepsis respectively. Multiple logistic regression models showed that maternal UTI during the third trimester of pregnancy was a significant predictor of both early (aOR = 2.78, 95% CI: 1.07–7.28; p<0.05) and late-onset (aOR = 5.48, 95% CI: 1.58–18.99; p<0.05) neonatal sepsis. Among neonatal factors, premature birth (aOR = 2.84, 95% CI:1.13–7.12; p<0.05) and APGAR score <7 at 5 minutes (aOR = 2.58, 95% CI: 1.04–6.39; p<0.05) were associated with increased risk of EONS (Table 3) while only male sex of neonate diminished the risk of LONS by 67% (aOR = 0.33, 95% CI: 0.13–0.88; p<0.05) compared to female neonate (Table 4).

## Causative organisms of neonatal sepsis

*Escherichia coli* was the most frequently isolated gram-negative organism from blood samples of suspected neonates of sepsis (40.7%) followed by *Klebsiella pneumoniae* (18.7%). These two organisms were the most common cause of both early and late-onset neonatal sepsis, though *Klebsiella pneumoniae* was more likely associated with LONS (10.7% of EONS vs 31.4% of

**Table 3. Bivariate and multiple logistic regression for risk factors of early-onset neonatal sepsis.**

| Variables | Category | Case, N (%) | Control, N (%) | cOR (95% CI) | aOR (95% CI) |
|---|---|---|---|---|---|
| Maternal age (year) | <18 | 13 (23.2) | 22 (19.0) | 0.99 (0.29–3.34) | |
| | 18 to 35 | 37 (66.1) | 84 (72.4) | 0.73 (0.25–2.17) | |
| | >35 | 6 (10.7) | 10 (8.6) | | |
| Maternal UTI during 3rd trimester | Yes | 13 (23.2) | 11 (9.5) | 2.89 (1.20–6.94)* | 2.78 (1.07–7.28)* |
| | No | 43 (76.8) | 105 (90.5) | | |
| Prolong labor | Yes | 11 (19.6) | 13 (11.2) | 1.94 (0.81–4.65) | |
| | No | 45 (80.4) | 103 (88.8) | | |
| Premature rupture of membrane | Yes | 15 (26.8) | 19 (16.4) | 1.87 (0.87–4.03) | |
| | No | 41 (73.2) | 97 (83.6) | | |
| Mode of delivery | Caesarean | 29 (51.8) | 39 (33.6) | 2.12 (1.11–4.06)* | 1.50 (0.69–3.25) |
| | Vaginal | 27 (48.2) | 77 (66.4) | | |
| Sex of neonate | Female | 29 (51.8) | 46 (39.7) | 1.63 (0.86–3.11) | |
| | Male | 27 (48.2) | 70 (60.3) | | |
| Prematurity | Yes (<37 weeks) | 27 (48.2) | 24 (20.7) | 3.57 (1.79–7.12)* | 2.84 (1.13–7.12)* |
| | No (≥37 weeks) | 29 (51.8) | 92 (79.3) | | |
| Weight at birth (gram) | Low, <2500 | 31 (55.4) | 33 (28.4) | 3.12 (1.61–6.06)* | 1.25 (0.50–3.16) |
| | Normal, ≥2500 | 25 (44.6) | 83 (71.6) | | |
| APGAR score at 5 minutes | <7 | 14 (25.0) | 13 (11.2) | 2.64 (1.15–6.09)* | 2.58 (1.04–6.39)* |
| | ≥7 | 42 (75.0) | 103 (88.8) | | |

N.B.

*p-value <0.05, CI: Confidence interval, cOR: crude odds ratio, aOR: Adjusted odds ratio.

**Table 4. Bivariate and multiple logistic regression for risk factors of late-onset neonatal sepsis.**

| Variables | Category | Case, N (%) | Control, N (%) | cOR (95% CI) | aOR (95% CI) |
|---|---|---|---|---|---|
| Maternal age (year) | <18 | 8 (22.9) | 17 (22.1) | 1.01 (0.29–3.45) | |
| | 18 to 35 | 20 (57.1) | 45 (58.4) | 0.95 (0.34–2.69) | |
| | >35 | 7 (20.0) | 15 (19.5) | | |
| Maternal UTI during 3rd trimester | Yes | 9 (25.7) | 7 (9.1) | 3.46 (1.17–10.25)* | 5.48 (1.58–18.99)* |
| | No | 26 (74.3) | 70 (90.9) | | |
| Prolong labor | Yes | 11 (31.4) | 12 (15.6) | 2.48 (0.97–6.37) | |
| | No | 24 (68.6) | 65 (84.4) | | |
| Premature rupture of membrane | Yes | 10 (28.6) | 11 (14.3) | 2.40 (0.91–6.35) | |
| | No | 25 (71.4) | 66 (85.7) | | |
| Mode of delivery | Caesarean | 16 (45.7) | 26 (33.8) | 1.65 (0.73–3.74) | |
| | Vaginal | 19 (54.3) | 51 (66.2) | | |
| Sex of neonate | Female | 19 (54.3) | 57 (74.0) | 0.42 (0.18–0.96)* | 0.33 (0.13–0.88)* |
| | Male | 16 (45.7) | 20 (26.0) | | |
| Prematurity | Yes (<37 weeks) | 15 (42.9) | 12 (15.6) | 4.06 (1.64–10.09)* | 2.34 (0.71–7.70) |
| | No (≥37 weeks) | 20 (57.1) | 65 (84.4) | | |
| Weight at birth (gram) | Low, <2500 | 18 (51.4) | 16 (20.8) | 4.04 (1.71–9.56)* | 2.19 (0.64–7.53) |
| | Normal, ≥2500 | 17 (48.6) | 61 (79.2) | | |
| APGAR score at 5 minutes | <7 | 14 (40.0) | 9 (11.7) | 5.04 (1.91–13.29)* | 2.10 (0.61–7.26) |
| | ≥7 | 21 (60.0) | 68 (88.3) | | |

N.B.

*p-value <0.05, CI: Confidence interval, cOR: crude odds ratio, aOR: Adjusted odds ratio.

**Table 5. Causative organisms of neonatal sepsis.**

| Organism | Total, N (%) | Mode of onset | | Gestational age | | Birth weight | |
|---|---|---|---|---|---|---|---|
| | | EONS, N (%) | LONS, N (%) | <37 weeks, N (%) | ≥37 weeks, N (%) | <2500 g, N (%) | ≥2500 g, N (%) |
| *Escherichia coli* | 37 (40.7) | 26 (46.4) | 11 (31.4) | 17 (40.5) | 20 (40.8) | 22 (44.9) | 15 (35.7) |
| *Klebsiella pneumoniae* | 17 (18.7) | 6 (10.7) | 11 (31.4) | 8 (19.0) | 9 (18.4) | 10 (20.4) | 7 (16.7) |
| *Staphylococcus aureus* | 25 (27.5) | 17 (30.4) | 8 (22.9) | 11 (26.2) | 14 (28.6) | 11 (22.4) | 14 (33.3) |
| *Staphylococcus saprophyticus* | 8 (8.8) | 5 (8.9) | 3 (8.6) | 5 (11.9) | 3 (6.1) | 4 (8.2) | 4 (9.5) |
| *Staphylococcus epidermidis* | 3 (3.3) | 1 (1.8) | 2 (5.7) | 1 (2.4) | 2 (4.1) | 2 (4.1) | 1 (2.4) |
| *Streptococcus viridans* | 1 (1.1) | 1 (1.8) | 0 (0.0) | 0 (0.0) | 1 (2.0) | 0 (0.0) | 1 (2.4) |

LONS). Among gram-positive organisms, *Staphylococcus aureus* (27.5%) and *Staphylococcus saprophyticus* (8.8%) were most commonly isolated from blood samples (Table 5).

## Antibiotic resistance pattern

Table 6 shows the pattern of antibiotic resistance of causative organisms of neonatal sepsis. Among the beta-lactam antibiotics used for gram-negative organisms, both *Escherichia coli* and *Klebsiella pneumoniae* showed the highest resistance to amoxicillin and clavulanic acid combination (57% and 70.6% respectively). Though *Klebsiella pneumoniae* was highly sensitive to imipenem and meropenem, *Escherichia coli* was moderately resistant to these drugs (32% and 46% respectively). These organisms were highly resistant to almost all non-β-lactum

**Table 6. Antibiotic resistance pattern (percentage of resistance) of causative organisms of neonatal sepsis.**

| Antibiotic | Organism | | | | |
|---|---|---|---|---|---|
| | *Escherichia coli* | *Klebsiella pneumoniae* | *Staphylococcus aureus* | *Staphylococcus saprophyticus* | *Staphylococcus epidrmidis* |
| | (n = 37), R% | (n = 17), R% | (n = 25), R% | (n = 8), R% | (n = 3), R% |
| MDR | 100 | 100 | 100 | 100 | 100 |
| β-lactam antibiotics | | | | | |
| Amoxicillin | | | 92.0 | 37.5 | 100 |
| Amoxicillin + clavulanic acid | 56.8 | 70.6 | 44.0 | 50.0 | 33.3 |
| Cloxacillin | | | 52.0 | 25.0 | 33.3 |
| Imipenem | 32.4 | 0.0 | 24.0 | 12.5 | 33.3 |
| Meropenem | 45.9 | 0.0 | 40.0 | 25.0 | 33.3 |
| Non-β-lactam antibiotics | | | | | |
| Ceftriaxone | 75.7 | 76.5 | | 62.5 | 66.7 |
| Ceftazidime | 62.2 | 58.8 | 64.0 | 100 | 100 |
| Cefuroxime | 89.2 | 64.7 | 60.0 | 62.5 | 0.0 |
| Cefepime | 73.0 | 47.1 | 48.0 | 25.0 | 33.3 |
| Azithromycin | | | 80.0 | 100 | 100 |
| Gentamicin | 75.7 | 64.7 | 48.0 | 75.0 | 33.3 |
| Amikacin | 70.3 | 64.7 | 32.0 | 75.0 | 66.7 |
| Ciprofloxacin | 78.4 | 52.9 | 56.0 | 75.0 | 33.3 |
| Levofloxacin | 40.5 | 35.3 | 56.0 | 50.0 | 0.0 |
| Cotrimoxazole | 73.0 | 76.5 | 72.0 | 75.0 | 66.7 |
| Doxycycline | 78.4 | 52.9 | 44.0 | 75.0 | 33.3 |
| Nitrofurantoin | 56.8 | 52.9 | 20.0 | 50.0 | 0.0 |
| Linezolid | | | 4.0 | 12.5 | 33.3 |

R = Resistance

antibiotics tested, such as the third-generation cephalosporins, aminoglycosides, and quinolones. However, levofloxacin was quite sensitive against these organisms (resistance rate 40% and 35% for *E. coli* and *K. pneumoniae* respectively).

Gram-positive organisms were moderately sensitive to almost all β-lactum antibiotics except amoxicillin. Resistance rates of these organisms to Non-β-lactum antibiotics like the third-generation cephalosporins, aminoglycosides, and quinolones, and others were comparatively higher than β-lactum antibiotics. However, amikacin, nitrofurantoin, and linezolid were still sensitive against the gram-positive organisms, especially *Staphylococcus aureus* (Table 6).

## Discussion

The present study aimed to evaluate the maternal and neonatal risk factors of developing sepsis, both early and late-onset, among the neonates delivered in a tertiary care hospital in northern Bangladesh. Among the diagnosed neonatal sepsis cases, 61.5% were diagnosed as EONS and 38.5% were diagnosed as LONS. The prevalence of EONS was much lower found in a study conducted in Dhaka, Bangladesh (35%) [27]; the sample size of that study was not large enough to conclude. Our finding is consistent with previous studies conducted in different countries of similar economic conditions like India (67%) [12], Nepal (78%) [10], Ethiopia (77%) [19], and Ghana (82%) [25].

Our study showed that neonates with a history of maternal UTI during the third trimester of pregnancy were five times more prone to developing sepsis. This was also found in the case of both EONS and LONS when the regression model was carried out separately. A similar result was found in a meta-analysis including studies from all over the world; it reported that lab-confirmed maternal infection or bacterial colonization significantly increases the risk of neonatal sepsis during the first week of life, which includes both EONS and LONS according to our definition [28]. Maternal infections may frequently transmit to the baby in utero or during passage through the birth canal which usually causes neonatal sepsis. Other maternal risk factors, prolonged labor and premature rupture of membrane, which increase the risk of the chance of ascending infection from the birth canal into the amniotic fluid were found to increase the risk of development of neonatal sepsis in different studies [19, 28, 29], though these factors were not associated with increased risk of sepsis in our study. Perhaps proper management of prolonged labor and premature rupture of the membrane in our tertiary care setting had minimized the risk of ascending infection.

Among neonatal factors, premature birth and APGAR scores of <7 at 5 minutes were found to be associated with overall neonatal sepsis as well as developing EONS. We could not find any association between low birth weight and neonatal sepsis. Preterm birth and low birth weight were reported as risk factors of neonatal sepsis in several previous studies [17, 18, 30, 31]. However, some evidence found no association between neonatal sepsis and preterm birth or low birth weight [19, 26]. A lower APGAR score, which leads to perinatal asphyxia resulting in immunological insult, was also reported as an important risk factor of neonatal sepsis in different studies [16, 19, 26].

In our study, gram-negative organisms were more commonly isolated than gram-positive organisms. *Escherichia coli* was the predominant gram-negative bacteria followed by *Klebsiella pneumoniae* whereas *Staphylococcus aureus* was the predominant gram-positive organism. These organisms were also most commonly isolated from the neonatal blood samples of a previous study in Bangladesh [17]. Another study from Bangladesh reported *Klebsiella pneumoniae* as the most commonly isolated organism [27]. However, this study included blood samples mostly from LONS patients, and *Klebsiella pneumoniae* mainly caused LONS also in our study samples. *Klebsiella pneumonia*, *Escherichia coli*, *Staphylococcus aureus*, and

coagulase-negative *Staphylococcus* were reported as the most common causative organisms of neonatal sepsis worldwide [9, 10, 12, 32]. *Escherichia coli* and *Klebsiella pneumoniae* are normal members of gastrointestinal flora and cause a range of infections including UTI, pneumonia, and septicemia. Maternal colonization by these organisms increases the neonatal risk of being infected by them [28].

Antimicrobial resistance is a major challenge for clinical management of neonatal sepsis. In our study, all the isolated bacteria were multidrug-resistant. Both gram-negative and gram-positive organisms were highly resistant to commonly available antibiotics like amoxicillin, cephalosporins, aminoglycosides, and quinolones. A similar pattern of a very high prevalence of multidrug-resistant organisms was also reported in previous studies from Bangladesh [17, 27] and also from different countries of the world like India, Nepal, and Ethiopia [10–12]. Our study revealed that the carbapenemase group of drugs (imipenem and meropenem) was mostly sensitive against both gram-positive and gram-negative bacteria. Some other antibiotics like amikacin, nitrofurantoin, and linezolid were also sensitive, especially against gram-positive organisms (*Staphylococcus aureus*). A similar pattern of sensitivity was also reported in previous studies conducted in Bangladesh [27] and the neighboring countries like India and Nepal [10, 12].

Despite being a tertiary care hospital, the study institution lacks specific protocol regarding the use of the first, second, and third-line antibiotics used to treat neonatal sepsis. Broad-spectrum antibiotics, mainly ceftriaxone and amikacin are being used most commonly as empirical antibiotics in case of suspected sepsis at the time of admission, as these two antibiotics are available in the hospital through government supply. However, ceftriaxone showed a high level of resistance, though amikacin remained moderately sensitive, mostly against gram-positive bacteria. Based on our findings, levofloxacin and imipenem combination may be the drug of choice for empirical use in neonatal sepsis.

## Strength and limitation of the study

The present study was the very first attempt to identify the risk factors and causative organisms of sepsis among hospital-borne neonates in the northern region of Bangladesh, to the best of our knowledge. Despite the fact, we would like to admit some limitations of the study. Firstly, this was a single-center study with a small sample size and both cases and controls were included from neonates admitted in the same tertiary care hospital. The findings may not be inferential for the neonates borne in the home or primary care settings. Secondly, we could not include detailed socio-demographic information due to our limited resources, which may have a role in developing neonatal sepsis. Thirdly, all the mothers classified as having UTI in the third trimester did not have proof of positive urine culture and in those cases, having >5 pus cells per HPF was considered as evidence of UTI. So there is a probability of classifying the mothers having sterile pyuria as UTI that can potentially confound the result. Finally, specific characterization of MDR bacteria (such as MRSA, ESBL, etc.) and molecular-based specifications have not been done due to a lack of well-established laboratory facilities, and detection of some UTIs might be missed to make an underestimated result. Therefore, the results of this study are recommended to be interpreted with caution. Further study with a larger sample of cases from multiple centers and healthy controls from the community is suggested to verify the risk factors found in our study. Other socio-demographic factors that we could not include due to our limited resources should also be considered. Moreover, the molecular characterization of MDR bacteria is suggested for a better understanding of the epidemiology and resistance mechanism of the causative organisms of neonatal sepsis.

## Conclusions

Our result has demonstrated that the risk of developing neonatal sepsis was higher during the first 72 hours of a neonate's life. Maternal history of UTI during the third trimester of pregnancy along with some neonatal factors such as premature birth and APGAR score of <7 at 5 minutes after birth was the significant risk factor of neonatal sepsis. Proper and adequate antenatal screening for early diagnosis and treatment of maternal infection during pregnancy as well as identifying high-risk pregnancy for adequate perinatal management of neonates are recommended to prevent neonatal sepsis-related morbidity and mortality. It was also observed that a high prevalence of multidrug-resistant organisms had made clinical management challenging. According to our findings, linezolid, nitrofurantoin, and carbapenems (meropenem and meropenem) may be the potential drugs of choice for empirical therapy to treat neonatal sepsis. However, the antibiotics should preferably be used according to the culture and sensitivity report at the earliest opportunity to reduce the risk of developing resistance against these drugs.

## Supporting information

**S1 Data.**
(SAV)

## Acknowledgments

The authors would like to acknowledge Md. Nurunnabi, lab assistant of Rajshahi Medical College Hospital for his support throughout the study. The authors would also like to express their sincere gratitude to all the parents of the study participants and the staff engaged in the study.

## Author Contributions

**Conceptualization:** Md. Abdur Rafi.

**Data curation:** Md. Abdur Rafi, M. Morsed Zaman Miah.

**Formal analysis:** Md. Abdur Rafi, M. Morsed Zaman Miah, Md. Golam Hossain.

**Investigation:** Md. Golam Hossain.

**Methodology:** M. Morsed Zaman Miah, Md. Abdul Wadood, Md. Golam Hossain.

**Resources:** Md. Abdul Wadood, Md. Golam Hossain.

**Supervision:** Md. Golam Hossain.

**Validation:** Md. Abdul Wadood, Md. Golam Hossain.

**Writing – original draft:** Md. Abdur Rafi.

**Writing – review & editing:** M. Morsed Zaman Miah, Md. Abdul Wadood, Md. Golam Hossain.

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
