## [Decision Letter · Decision Letter 0]

4 Sep 2020

PONE-D-20-13278

Risk factors and etiology of neonatal sepsis after hospital delivery: a case-control study in a tertiary care hospital of Rajshahi, Bangladesh

PLOS ONE

Dear Dr. Hossain,

Thank you for submitting your manuscript to PLOS ONE. After careful consideration, we feel that it has merit but does not fully meet PLOS ONE’s publication criteria as it currently stands. Therefore, we invite you to submit a revised version of the manuscript that addresses the points raised during the review process.

The manuscript has been evaluated by two reviewers, their comments are available below.

The reviewers have raised a number of concerns, regarding the reporting of methodological aspects of the study. In addition, they request a careful consideration of the discussion to ensure that the possible limitations are adequately discussed. Please also note that your manuscript requires copyediting.

Could you please carefully revise the manuscript to address all comments raised.

We look forward to receiving your revised manuscript.

Kind regards,

Sara Fuentes Perez, PhD

Staff Editor

PLOS ONE

Journal Requirements:

Reviewers' comments:

Reviewer's Responses to Questions

**Comments to the Author**

1. Is the manuscript technically sound, and do the data support the conclusions?

Reviewer #1: Yes

Reviewer #2: Yes

2. Has the statistical analysis been performed appropriately and rigorously? 

Reviewer #1: Yes

Reviewer #2: Yes

3. Have the authors made all data underlying the findings in their manuscript fully available?

Reviewer #1: Yes

Reviewer #2: Yes

4. Is the manuscript presented in an intelligible fashion and written in standard English?

Reviewer #1: Yes

Reviewer #2: Yes

5. Review Comments to the Author

Reviewer #1: The study evaluates risk factors and etiology of neonatal sepsis, is well designed, but there is a major point needing to be clarified

MAJOR COMMENTS

Diagnosis of neonatal sepsis was based on positive blood culture, in this sense is important to describe the sensitivity and specificity of the of the diagnosis assay.

Although Blood culture assays is considered a gold standard, it has showed limited sensitivity for neonatal sepsis diagnosis, especially in low birth weight premature neonates in who its difficult to draw 1 ml blood due to their weight and total blood body volume; how does this explain?

Table 5 show a descriptive analysis, it would be desirable to include bivariate and multiregression analysis if applicable for causal bacteria and EONS or LONS sepsis

In the multidrug-resistant analysis authors report high prevalence of MDR multidrug-organisms; however, the MDR was no characterized on molecular based assay as the authors describe in the limitations statement. Therefore, its recommends to take the results with caution. Please explain the concern and resolve.

MINOR COMMENTS

Mistakes in the bibliographic references

There are some mistakes in the bibliography; in the introduction in page 5 line 94 reference appear as “?”. Please check and resolve.

Is desirable to refer the international consensus for definition of sepsis: Goldstein B, Giroir B, Randolph A. International pediatric sepsis consensus conference: Definitions for sepsis and organ dysfunction in pediatrics. Pediatr Crit Care Med 2005; 6: 2-8.

Reviewer #2: 1)Profreading preferably by a native speaker

2)Introduction:INTRODUCTION :line 95-96:the cited study is not clear,there is no relation with the sentence just before and after it.

METHODS: Line 114 Expectant mothers are diagnosed to have UTI based on wcc count in the urine .As cultures are the gold standard to diagnose UTI ,would you regard everyone with sterile pyuria to have UTI?

121:IMCI should be replace with IMNCI

124:What was the indication of admission for control neonates

Discussion:Please mention the protocol regarding the use of first ,second and third line antibiotics used to treat neonatal sepsis in your centre,if any.If no any such protocol what is the current most common empricial antibiotic used in a case of suspected sepsis at the time of admission?

6. PLOS authors have the option to publish the peer review history of their article (what does this mean?). If published, this will include your full peer review and any attached files.

Reviewer #1: No

Reviewer #2: **Yes: **Dr Bhishma Pokhrel

---

## [Author Response · Author response to Decision Letter 0]

1 Oct 2020

Journal Name: PLOS ONE 

Tracking No. (Manuscript ID): PONE-D-20-13278

Manuscript Title: " Risk factors and etiology of neonatal sepsis after hospital delivery: a case-control study in a tertiary care hospital of Rajshahi, Bangladesh" 

Dear Editor

Thank you very much for providing reviewer’s comments on our manuscript. We have modified and revised the manuscript accordingly, and detailed point–by-point corrections are given below:

Response to reviewer comments:

Reviewer #1: The study evaluates risk factors and etiology of neonatal sepsis, is well designed, but there is a major point needing to be clarified. 

Response to Reviewer Comments: Thank you very much for your comments on our manuscript. 

MAJOR COMMENTS

Diagnosis of neonatal sepsis was based on positive blood culture, in this sense is important to describe the sensitivity and specificity of the of the diagnosis assay.

Response to Reviewer Comments: We have described this issue with reference in Page 8 Line 158-159. [Manuscript with Track Changes]. 

Although Blood culture assays is considered a gold standard, it has showed limited sensitivity for neonatal sepsis diagnosis, especially in low birth weight premature neonates in who its difficult to draw 1 ml blood due to their weight and total blood body volume; how does this explain?

Response to Reviewer Comments: We have described and explained the issue in Page 8, Line 152-157. [Manuscript with Track Changes]. 

Table 5 show a descriptive analysis, it would be desirable to include bivariate and multi regression analysis if applicable for causal bacteria and EONS or LONS sepsis

Response to Reviewer Comments: Thank you very much for your comments, Table 5 shows the frequency of each category of mode of onset, gestational age and birth weight for different organisms such as Escherichia coli, Klebsiella pneumoniae, etc. So, we think there are no scope to apply bivariate and multiple regression analysis in here. 

In the multidrug-resistant analysis authors report high prevalence of MDR multidrug-organisms; however, the MDR was no characterized on molecular based assay as the authors describe in the limitations statement. Therefore, its recommends to take the results with caution. Please explain the concern and resolve.

Response to Reviewer Comments: The concern was briefly discussed in Page 21, Line 342-344. [Manuscript with Track Changes]

MINOR COMMENTS

Mistakes in the bibliographic references

There are some mistakes in the bibliography; in the introduction in page 5 line 94 reference appear as “?”. Please check and resolve.

Response to Reviewer Comments: We have checked and made correction [Page 5 Line 94]. [Manuscript with Track Changes]

Is desirable to refer the international consensus for definition of sepsis: Goldstein B, Giroir B, Randolph A. International pediatric sepsis consensus conference: Definitions for sepsis and organ dysfunction in pediatrics. Pediatr Crit Care Med 2005; 6: 2-8.

Response to Reviewer Comments: According to your suggestions, we have revised this issue in Page 7-8 Line 138-146. [Manuscript with Track Changes]

Reviewer #2: 

1) Prof-reading preferably by a native speaker

Response to Reviewer Comments: Thank you for your comments on our manuscript. We have tried with our best to make correction in English throughout the manuscript.

2) Introduction: INTRODUCTION : line 95-96:the cited study is not clear, there is no relation with the sentence just before and after it.

Response to Reviewer Comments: We have made correction [Page 5 Line 94]. [Manuscript with Track Changes]

METHODS: Line 114 Expectant mothers are diagnosed to have UTI based on wcc count in the urine .As cultures are the gold standard to diagnose UTI ,would you regard everyone with sterile pyuria to have UTI?

Response to Reviewer Comments: We described this issue in Page 6 Line 115-118 and in Page 21, Line 337-340. [Manuscript with Track Changes]

121: IMCI should be replace with IMNCI

Response to Reviewer Comments: We have made correction [Page 7 Line 125]. [Manuscript with Track Changes]

124:What was the indication of admission for control neonates

Response to Reviewer Comments: The indication of admission for control neonates has been mentioned in Page 7 Line 128-129. [Manuscript with Track Changes]

Discussion: Please mention the protocol regarding the use of first, second and third line antibiotics used to treat neonatal sepsis in your centre, if any. If no any such protocol what is the current most common empricial antibiotic used in a case of suspected sepsis at the time of admission?

Response to Reviewer Comments: We have discussed about this issue in Page 20 Line 324-328. [Manuscript with Track Changes]

We would like to thank the reviewers for the valuable comments. We have revised the documents to the best of our ability, but we will definitely be happy to provide further improvement if there are further clarifications required. 

With best regards

Dr. Md. Golam Hossain

Professor of Health Research Group

Department of Statistics, University of Rajshahi

Rajshahi-6205, Bangladesh

E-mail: hossain95@yahoo.com

---

## [Decision Letter · Decision Letter 1]

22 Oct 2020

PONE-D-20-13278R1

Risk factors and etiology of neonatal sepsis after hospital delivery: a case-control study in a tertiary care hospital of Rajshahi, Bangladesh

PLOS ONE

Dear Dr. Hossain,

Thank you for submitting your manuscript to PLOS ONE. After careful consideration, we feel that it has merit but does not fully meet PLOS ONE’s publication criteria as it currently stands. Therefore, we invite you to submit a revised version of the manuscript that addresses the points raised during the review process.

Your revised manuscript has been re-reviewed by the original reviewers. The reviewer still has questions. Also, I would like to suggest that you ask an Native English speaker to go through your manuscript and makes the necessary edition before your re-submission. 

We look forward to receiving your revised manuscript.

Kind regards,

Yung-Fu Chang

Academic Editor

PLOS ONE

Reviewers' comments:

Reviewer's Responses to Questions

**Comments to the Author**

1. If the authors have adequately addressed your comments raised in a previous round of review and you feel that this manuscript is now acceptable for publication, you may indicate that here to bypass the “Comments to the Author” section, enter your conflict of interest statement in the “Confidential to Editor” section, and submit your "Accept" recommendation.

Reviewer #1: All comments have been addressed

Reviewer #2: All comments have been addressed

2. Is the manuscript technically sound, and do the data support the conclusions?

Reviewer #1: Yes

Reviewer #2: Yes

3. Has the statistical analysis been performed appropriately and rigorously? 

Reviewer #1: Yes

Reviewer #2: Yes

4. Have the authors made all data underlying the findings in their manuscript fully available?

Reviewer #1: Yes

Reviewer #2: Yes

5. Is the manuscript presented in an intelligible fashion and written in standard English?

Reviewer #1: Yes

Reviewer #2: Yes

6. Review Comments to the Author

Reviewer #1: The authors attended all of the recommendations; corrections were made in the background, methodology, results and bibliography sections.

The manuscript is OK.

Reviewer #2: There are still some errors in english language at some places so please try to be as accurate as possible .

What combination of two antibiotics would you recommend to start as an empirical therapy in a case of suspected sepsis as concluded from this study as opposed to the existing ceftriaxone and amikacin ?

7. PLOS authors have the option to publish the peer review history of their article (what does this mean?). If published, this will include your full peer review and any attached files.

Reviewer #1: **Yes: **Yelda A. Leal

Reviewer #2: **Yes: **Bhishma Pokhrel

---

## [Author Response · Author response to Decision Letter 1]

27 Oct 2020

Journal Name: PLOS ONE 

Manuscript ID: PONE-D-20-13278R1

Manuscript Title: “Risk factors and etiology of neonatal sepsis after hospital delivery: a case-control study in a tertiary care hospital of Rajshahi, Bangladesh”

Dear Editor

Thank you very much for providing reviewer’s comments on our manuscript. We have modified and revised the manuscript accordingly, and detailed point–by-point corrections are given below:

Response to reviewer comments:

Reviewer #2: 

There are still some errors in English language at some places so please try to be as accurate as possible.

Response to Reviewer Comments: Thank you very much for your comments on our manuscript. We have revised and tried our best to improve English language of the manuscript. 

What combination of two antibiotics would you recommend to start as an empirical therapy in a case of suspected sepsis as concluded from this study as opposed to the existing ceftriaxone and amikacin?

Response to Reviewer Comments: We have discussed about this issue in Page, 20; Line: 328-329. 

We would like to thank the reviewers for the valuable comments. We have revised the documents to the best of our ability, but we will definitely be happy to provide further improvement if there are further clarifications required. 

With best regards

Dr. Md. Golam Hossain

Professor of Health Research Group

Department of Statistics, University of Rajshahi

Rajshahi-6205, Bangladesh

E-mail: hossain95@yahoo.com

---

## [Decision Letter · Decision Letter 2]

30 Oct 2020

Risk factors and etiology of neonatal sepsis after hospital delivery: a case-control study in a tertiary care hospital of Rajshahi, Bangladesh

PONE-D-20-13278R2

Dear Dr. Hossain,

We’re pleased to inform you that your manuscript has been judged scientifically suitable for publication and will be formally accepted for publication once it meets all outstanding technical requirements.

Kind regards,

Yung-Fu Chang

Academic Editor

PLOS ONE

Additional Editor Comments (optional):

Reviewers' comments:

Reviewer's Responses to Questions

**Comments to the Author**

1. If the authors have adequately addressed your comments raised in a previous round of review and you feel that this manuscript is now acceptable for publication, you may indicate that here to bypass the “Comments to the Author” section, enter your conflict of interest statement in the “Confidential to Editor” section, and submit your "Accept" recommendation.

Reviewer #2: All comments have been addressed

2. Is the manuscript technically sound, and do the data support the conclusions?

Reviewer #2: Yes

3. Has the statistical analysis been performed appropriately and rigorously? 

Reviewer #2: Yes

4. Have the authors made all data underlying the findings in their manuscript fully available?

Reviewer #2: Yes

5. Is the manuscript presented in an intelligible fashion and written in standard English?

Reviewer #2: Yes

6. Review Comments to the Author

Reviewer #2: minor grammatical errors like line 357" was the significant risk factor of neonatal sepsis" should be replaced by were ,line 362/363 has typing mistakes meropenem is repeated twice ,please repace one with imipenem.

7. PLOS authors have the option to publish the peer review history of their article (what does this mean?). If published, this will include your full peer review and any attached files.

Reviewer #2: **Yes: **Bhishma Pokhrel

---

## [Editor Report · Acceptance letter]

5 Nov 2020

PONE-D-20-13278R2 

Risk factors and etiology of neonatal sepsis after hospital delivery: a case-control study in a tertiary care hospital of Rajshahi, Bangladesh 

Dear Dr. Hossain:

I'm pleased to inform you that your manuscript has been deemed suitable for publication in PLOS ONE. Congratulations! Your manuscript is now with our production department. 

Kind regards, 

on behalf of

Dr. Yung-Fu Chang 

Academic Editor

PLOS ONE